

# Historic checklist, core species and temporal composition turnover of birds in an urban protected natural area in central México over 50 years

Ubaldo Márquez-Luna[1], Pablo Arenas[2], M. Isabel Herrera-Juárez[3], Hilda Marcela Pérez-Escobedo[2], Gustavo Hernández-Orta[4] and Guillermo Gil-Alarcón[2]

[1] Laboratorio de Ecología, UBIPRO, Facultad de Estudios Superiores-Iztacala, Universidad Nacional Autónoma de México, Tlalnepantla de Baz, Estado de México, Mexico
[2] Secretaría Ejecutiva de la Reserva Ecológica del Pedregal de San Ángel, Ciudad Universitaria, Universidad Nacional Autónoma de México, Coyoacán, Ciudad de México, Mexico
[3] Coordinación de Estrategias para la Biodiversidad, Dirección General de Coordinación de Políticas y Cultura Ambiental de la Secretaría del Medio Ambiente del Gobierno de la Ciudad de México, Cuauhtémoc, Ciudad de México, Mexico
[4] Tótotl, Aves y Medio Ambiente, A. C., Álvaro Obregón, Ciudad de México, Mexico

Corresponding author
Ubaldo Márquez-Luna,
marquezubaldo@gmail.com

## ABSTRACT

The Pedregal de San Ángel Ecological Reserve (REPSA) is located within the central campus of the National Autonomous University of Mexico (UNAM), in southwestern Mexico City. This area is known for its bird richness, which has been documented over time. However, this historical information has not been compiled, nor has it been assessed whether species composition has changed over time. In this study, we generated the historical checklist of avifauna in the REPSA by integrating data from systematic monitoring, bibliographic review and citizen science. Additionally, we calculated Jaccard's dissimilarity Index (nestedness and species turnover) among bird checklists from each year and evaluated whether this index and its components changed over time. In addition, we estimate whether the proportion of species arriving and becoming extinct in the REPSA is influenced by time (census interval) and by the migratory status of the species. Finally, we identified the core species of the REPSA. Our results indicate that from 1967 to 2023, 258 bird species have been recorded in the REPSA. The Jaccard's dissimilarity and turnover increased and reached an asymptotic trend over time while nestedness showed similar values throughout time. The time and species residency status did not have significant effect on the proportion of species that arrival and become extinct in the reserve. We identified 58 core species based on their persistence; 35 of these are residents, 22 have some migratory movement and one exotic species. Our results highlight the importance of the REPSA as a fundamental habitat for the avifauna of Mexico City, including a wide variety of migratory bird species.

## INTRODUCTION

The Pedregal de San Ángel Ecological Reserve (REPSA) is a protected natural area located within the central campus of the National Autonomous University of Mexico (UNAM), in southwestern Mexico City. This area has a great richness of birds that has been documented in several research studies (*Arizmendi, Espinosa & Ornelas, 1994*; *Chávez & Gurrola, 2007*, *2009*). Additionally, areas within the REPSA are frequently visited by birders participating in citizen science initiatives such as eBird and iNaturalist. However, historical information on bird richness has not been compiled or integrated, nor has it been assessed whether species composition within the reserve has changed over time. It is important to address this information gap due to the pressures on this habitat from the urbanization of Mexico City and the university campus. Urbanization processes are associated with human population growth and tend to reduce the number and quality of native vegetation patches, increase impervious surfaces and facilitate the introduction and establishment of exotic species (*Hansen et al., 2005*). During the last decade (2010–2020) the human population in Mexico City reached 9,209,944 inhabitants (*INEGI, 2020*). This population increase places greater pressure on the limits of natural areas due to the construction of urban infrastructure and demand for resources. Over time, within UNAM's central campus, infrastructure construction has increased landscape fragmentation and reduced the number of native vegetation patches in the area (*Peralta & Prado, 2009*). These disturbances could impact the composition and ecological dynamics of the avifauna inhabiting the REPSA.

Species turnover is the change in the composition of a community as a result of species immigration and extinction processes (*Russell et al., 1995*). Migratory species have been reported to be an important component of the avifauna of Mexico City (*Meléndez-Herrada, Gómez de Silva & Ortega-Álvarez, 2016*). This pattern is evident in the REPSA; the latest avifauna checklist reported that 43% of the bird species present have some degree of migratory movement (*Chávez & Gurrola, 2009*). This flow of species during the fall and winter is generated by the convergence of Central and Eastern flyways that are interrelated in a larger migratory system (*La Sorte et al., 2014*). This migratory system crosses the Gulf and Central Mexico where REPSA is located. On the other hand, there are several factors associated with species extinction (*e.g.*, habitat loss, establishment and spread of exotic species, predation). However, these factors could affect species differentially. Evaluating both processes over time is essential to understanding the dynamics and role of migratory and resident species in the richness of birds in the REPSA, it is essential to determine whether this flow of species is persistent over time—in other words, whether the same migratory and resident species have continued to be found in the REPSA year after year.

Evaluating species' persistence over time will allow us to identify the area's core species, *i.e.*, species that persist over time and tend to be abundant, as opposed to transient species that occur intermittently and in low abundance (*Magurran & Henderson, 2003*; *Coyle, Hurlbert & White, 2013*). In general, the relationship between species and their permanence in an area is defined by the affinity of that species for the conditions and resources the area offers. A species that can obtain all the resources and conditions

necessary for its survival in an area will tend to remain in that area over time (*Magurran & Henderson, 2003*).

Given the seasonal dynamics of the bird community in the REPSA and the increasing urbanization of the landscape matrix in which it is immersed, we expected the bird species composition to have changed over time. *Russell et al. (1995)* reported that the expected relationship between turnover and time (census interval) is non-linear and asymptotic. Species turnover at shorter time scales tend to increase, while at longer time scales turnover becomes asymptotic (*Russell et al., 1995*). In this study, we examine the trend of change in the dissimilarity (and its components) of the REPSA's avifauna over 50 years. We expected that the turnover between species listings increases and reach an asymptotic trend with time (census interval). On the other hand, we expected the number of species recorded to be similar over time if there is a balance between the proportion of species that become extinct and those that reach the reserve. In addition, we estimate whether the proportion of species arriving and becoming extinct in the REPSA is influenced by time (census interval) and by the residency status of the species. Finally, we identified the core species of the REPSA based on their persistence. To explore these hypotheses, we set the following objectives: (1) generate a historical checklist of bird species recorded in the REPSA, (2) calculate Jaccard's dissimilarity index and its components between the annual listings of species to determine if there is any trend of change over time, (3) determine the proportion of species arriving and becoming extinct from the REPSA between each pair of species listings and assess whether this is influenced by time (census interval) and the migratory status of the birds, and (4) identify core species in the reserve (persistent over time). In order to meet these objectives, we integrated the information on bird species richness reported in the REPSA over the time using three sources of information: (1) monthly monitoring of avifauna in six representative areas of the reserve, carried out from December 2013 to December 2023; (2) intensive search for species listings published in theses, scientific articles, or book chapters; and (3) information gathered through citizen science databases.

## MATERIALS AND METHODS

### Study area

The REPSA is contained within the central campus of the National Autonomous University of Mexico (UNAM), in the southwest of Mexico City, between 19°18′31″–19°19′17″ North latitude and 99°10′20″–99°11′52″ West latitude. Its elevation range is between 2,200 and 2,277 m asl (*Castillo-Argüero et al., 2007*). Its climate is temperate sub-humid, with a rainy season from June to October. The average annual temperature is 15 °C, and the annual rainfall is 833 mm. It has three core zones covering 171.3 ha and 13 buffer zones covering 66 ha, which together protect a total of 237.3 ha. The reserve protects a unique vegetation community called xerophilous scrubland of "Palo Loco" (*Pittocaulon praecox*; *Rzedowski, 1954*). Within the reserve there are areas of native vegetation in different degrees of conservation and artificially modified areas (Botanical Garden and Cantera Oriente) that contain a variety of plant associations, including patches of coniferous forest, cacti, aquatic vegetation and exotic plant species. The volcanic origin of

the landscape within the reserve gives rise to described seven microenvironments: plains, cavities, crevices, hollows, promontories, walls and caves (*Castillo-Argüero et al., 2007*). There are even bodies of water within the Botanical Garden and the Cantera Oriente, a feature that is not present in the native areas of the reserve. Together, these features generate a highly heterogeneous environment in the reserve, which facilitates the establishment and presence of a large number of species.

## Bird monitoring

We selected eight of the 16 areas that make up the reserve and established six routes. Route 1 included the Botanical Garden, Zona Núcleo Poniente (ZNP) and zone A11; Route 2, the Espacio Escultórico (A12); Route 3, zone A13; Route 4, the Paseo de las Esculturas (A5); Route 5, the Zona Núcleo Suroriente (ZNSO), and Route 6, the Cantera Oriente (A3) (Fig. 1). We conducted 96 bird samplings from December 2013 to December 2023 in these six areas of the REPSA. These zones were selected because together they encompass the variety of resources and conditions that the reserve provides. The length of each route was determined based on the total surface and the existing trails in each zone (1 = 4.1 km; 2 = 0.82 km; 3 = 0.79 km; 4 = 0.86 km; 5 = 1 km and 6 = 2 km). We performed monthly monitoring from December 2013 to February 2020 and from August 2021 to December 2023. We alternated the zone in which each monitoring was carried out, visiting each study zone twice per year—once during the rainy season (June–October) and once during the dry season (November–May). Due to the health contingency caused by the Sars-CoV-2 virus, we suspended this monitoring between March 2020 and July 2021. All monitoring started between 7:00 and 8:00 h and finished between 11:00 and 12:00 h depending on the time of sunrise. This period was selected because it includes the hours of peak bird activity (*Bibby et al., 2000*). During monitoring we determined to the species level the birds we sighted or heard and recorded the number of individuals observed per species. Observations were made using 8 × 42 and 10 × 50 binoculars and the species were determined using specialized field guides (*Howell & Webb, 1995*; *Van Perlo, 2006*).

## Residence pattern and abundance

We determined the residency pattern of the bird species within the reserve based on the presence of each species during the monthly monitoring. We established the following residency categories: resident (R) = species that were present in the reserve throughout the year; winter migrant (MI) = species that were present in the reserve during fall and winter (September to February); summer migrant (MV) = species that were present in the reserve during spring and summer (March to August); transient migrant (TM) = species that uses the reserve as pass on their migratory route; escapes (E) = species outside their geographical range and that are legally or illegally traded, and whose presence in the reserve is likely due to accidental or deliberate releases. If a species did not have sufficient information to determine its residency pattern within the reserve, then we assigned it the main residency category for Mexico reported by *Berlanga et al. (2015)*.

We categorized the abundance of each species based on the total number of individuals observed over the 8 years of monitoring (2014–2019 and 2022–2023). We established the

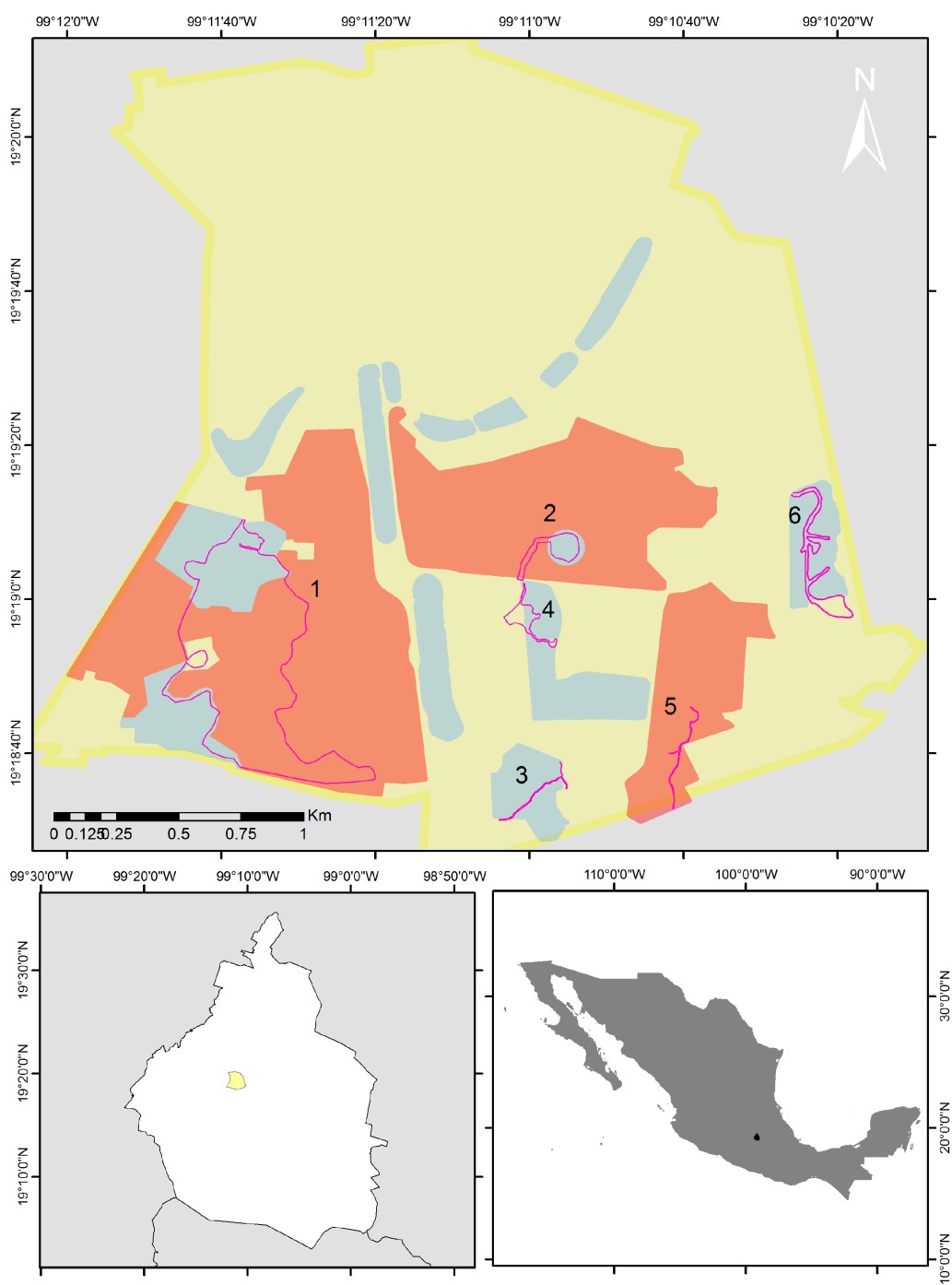

**Figure 1 Map of the study area.** Bird sampling routes within the Reserva del Pedregal de San Ángel (pink lines). The yellow polygon is the limit of Ciudad Universitaria (UNAM's central campus), red polygons are core areas and blue polygons are buffer areas of the reserve. Numbers indicate each of the following routes: 1 = the Botanical Garden, Zona Núcleo Poniente (ZNP) and zone A11; 2 = the Espacio Escultórico (A12); 3 = zone A13; 4 = the Paseo de las Esculturas (A5); 5 = the Zona Núcleo Suroriente (ZNSO), and 6 = the Cantera Oriente (A3). Map silhouttes: (© INEGI; https://www.inegi.org.mx/app/biblioteca/ficha.html?upc=794551091968).

following abundance categories: very abundant (VA) = species with more than 200 records; abundant (A) = species with 60–200 records; common (C) = species with 13–60 records, rare (R) = species with 4–12 records, very rare (VR) = species with 1–2 records. Species that were detected from sources other than our monitoring (see following section) were assigned the abundance reported in the source of information that recorded the species.

## Bibliographic review

We conducted a search for published scientific journal articles, dissertations or book chapters that reported a list of bird species present in any of the areas of the REPSA or Ciudad Universitaria (the UNAM campus), either as their main objective or as part of their results. The intensive search was conducted using the Scientific Electronic Library Online (SciELO), Searchable Ornithological Research Archive (SORA), Google Scholar and TESIUNAM (the online repository of UNAM dissertations). We used the following search terms and their combinations in Spanish and English: birds, avifauna, diversity, Ciudad Universitaria, UNAM, Reserva Ecológica, Pedregal, REPSA, xerophytic scrub, Jardín Botánico and Cantera Oriente. To prevent duplicity of data, only the first research published by each of the authors was taken into account. Additionally, we included bird records reported on the citizen science platforms iNaturalist (https://mexico.inaturalist.org) and eBird (https://ebird.org/home), we considered research-grade records (*i.e.*, there is consensus on the identification of the species) and for both databases we corroborated the correct identification of species in all records that included photographic support.

## Criteria for inclusion in the avifauna historic checklist

We established five criteria for including species records in the historical checklist of birds reported in REPSA; for a new species (*i.e.*, one not recorded in our monitoring) to be included in the historical checklist of birds reported in the REPSA, it had to meet criterion "1" or at least two of the other criteria (2–5). The criteria were: (1) the author of the report presented evidence confirming the record, consisting of photographs, audio or video recording, or detailed description of the species; (2) the species is known to be legally or illegally traded as an ornamental bird; (3) the geographic distribution of the species includes the reserve or central Mexico; (4) the species was recorded by at least two different authors; (5) the plumage coloration pattern of the species is unmistakable (*i.e.*, there is no other species reported in the area that has a similar plumage coloration pattern with which it could be confused).

## Dissimilarity in avifauna over time

To assess whether the composition of bird species present in the REPSA has changed over time, we estimated the dissimilarity between each pair of avifauna listing in the REPSA. In this analysis we included only studies that systematically sampled the REPSA avifauna during a full annual cycle. Citizen science database records were not included in this analysis because they were not obtained through systematic sampling. We used species incidence data (presence/absence) to calculate Jaccard's dissimilarity index and its two

components: dissimilarity due to species turnover (*i.e.*, the replacement of a species in one annual list by a different species in another list) and dissimilarity due to the difference in total species richness or nestedness (*i.e.*, number of total species recorded in a year due to the gain or loss of species in only one of the lists). Total dissimilarity and its two components take values that range from 0 to 1, with values closer to one representing stronger dissimilarity between annual lists overall, due to turnover, or due to nestedness, respectively (*Baselga & Orme, 2012*). We calculated Jaccard's dissimilarity and its components (species turnover and nestedness) using the betapart package (*Baselga & Orme, 2012*).

## Extinction and arrival of species

We determined the number of species that became extinct and arrived in the reserve between each pair of listings over time. We consider a species to be extinct in the reserve if its presence was recorded in the oldest listing and it was not recorded in the most recent listing with which it was compared. On the other hand, a species was considered to have arrived in the reserve when it was present in the most current listing and was not present in the oldest listing of the comparison. Each species that became extinct or arrived in the reserve was categorized based on its residency status. For this analysis we only used the following two residency categories: resident (species present in the reserve year-round) and migratory (*i.e.*, all species that make some migratory movement throughout the year). We exclude from this analysis the species categorized as escapes (E). Using these data we calculated the proportion of species that became extinct (resident and migratory) and that reached the reserve (resident and migratory) over time.

## Core and transient species

We determined the core and transient species based on the persistence of each species within the REPSA. To perform this classification, we determined the incidence (presence or absence) of each species in the reserve over time based on the lists reported in: 1) the research article compiled during the bibliographic review and 2) the data collected in our monitoring from 2014–2019 and 2022–2023. We only considered the sources of information that carried out a full annual cycle of avifauna sampling within the REPSA in order to ensure that sampling effort was equivalent and that migratory species were included as core species if their presence has been persistent over time within the reserve. A species was considered a core species if was recorded in at least two-thirds (>66% persistence) of the species lists analyzed (*Coyle, Hurlbert & White, 2013*). We categorized transient species as those that were recorded in less than one third of the listings (<33% persistence). Species that had persistence greater than 33% and less than 66% were considered as species with intermediate persistence. In addition, we determined whether species persistence was related to their abundance in the REPSA.

## Statistical analysis

We used GLMs (binomial distribution and logit link function) to tests whether proportion of species arriving and becoming extinct in the REPSA is influenced by time (census

interval) and the migratory status of the species. In these GLMs, we included the census interval and the residency status (resident and migrant) as explanatory variables and the proportion of species that arrival and becoming extinct as response variables. We tested two binomial response variables (1) extinct species (resident and migratory) and (2) species arrival (resident and migratory). Both response variables were calculated considering the total number of species that became extinct or reached the reserve when comparing two species listings. Subsequently, the proportion of resident and migratory species that became extinct or arrived in the reserve was determined. In all models the interactions among variables were tested. Statistical analyses were performed using R software v.3.5.3 (R Core Team, 2019), and the results were plotted using the ggplot2 package v. 3.1.0 (Wickham, 2016) and treemap v. 2.4-4 (Tennekes & Ellis, 2017).

## RESULTS

As result of bird monitoring we recorded the presence of 157 bird species. As a result of the bibliographic review, we found 14 published studies, of which seven were undergraduate theses in biology, five were book chapters and two were research articles (Table 1). The time range covered by these studies was from 1973 (Ramos, 1974) to 2019 (Rodríguez, 2020). Together, these publications contributed 52 species that were not recorded by any other source of information (Table 1). In the citizen science platforms, the oldest record of avifauna in the study area was from 2005 in iNaturalist and 1967 in eBird. Together, the two databases provided 49 species that were not recorded by any other source of information (Table 1). Integrating the results obtained through our monitoring, bibliographic review and data from citizen science initiatives from 1967 to 2023 within the REPSA resulted in records of a total of 258 bird species belonging to 18 Orders and 50 families (following the taxonomy of Chesser et al., 2023, Table 2).

The best represented families were Parulidae and Tyrannidae, with 29 and 26 species respectively. Based on our data of residency within the reserve, we determined that 55 species are residents, 66 are winter migrants, six are summer migrants, 16 species are considered escapes and 115 species did not have sufficient information to determine its residency pattern within the reserve. However, the categories of residence of these species for Mexico are as follows: 68 residents, 38 winter migrants, two are summer migrants and seven are transient migrants (Berlanga et al., 2015). Considering the total number of bird records (10,882 records) obtained through our avifauna monitoring between December 2013 and December 2023, we classified 15 species as very abundant, 22 as abundant, 40 as common, 42 as rare and 139 as very rare (Table 2, Supplemental Information). The resident species with the highest number of records was the House Finch (*Haemorhous mexicanus*), while the migratory species with the highest number of records was the Yellow-rumped Warbler (*Setophaga coronata*).

The lowest Jaccard's dissimilarity values occurred at low census intervals (1 to 5 years), while the highest dissimilarity values were reached at census intervals greater than 30 years (Fig. 2, blue dots and blue line). The species turnover shows a pattern similar to Jaccard's dissimilarity, with the lowest turnover values occurring between low census intervals (1 to 5 years) and the highest turnover occurring at census intervals greater than 20 years (Fig. 2,

**Table 1 Details of the sources of information used to construct the historical checklist of bird species reported in the REPSA.** It is indicated the type of study conducted, its duration, the areas of the reserve evaluated, the total number of bird species reported, and the number of species included in the reserve's historical checklist are reported. Reserve areas: JB = Botanical Garden, A3 = Cantera Oriente, ZNP = Zona Núcleo Poniente, A12 = Espacio Escultórico, A5 = Paseo de las Esculturas, ZNSO = Zona Núcleo Suroriente, Scrubland = areas with native vegetation without specifying the study area prior to the reserve's delimitation, REPSA = refers to areas with native vegetation without specifying the study area within the reserve. Data bases: iNaturalist (https://mexico.inaturalist.org/observations?order=asc&place_id=112247&subview=map&taxon_id=3) and eBird (https://ebird.org/hotspots?env.minX=-99.3267509999999&env.minY=19.028231&env.maxX=-98.8529829999999&env.maxY=19.53966).

| Author | Study type | Duration of the study | Studied area | Total species | Species in checklist |
|---|---|---|---|---|---|
| *Ramos (1974)* | Thesis | August 1972–June 1973 | JB | 73 | 4 |
| | | | Scrubland | 39 | |
| *González (1984)* | Thesis | December 1980–February 1982 | JB | 57 | 6 |
| *Álvarez-Sánchez et al. (1994)* | Book chapter | August 1972–June 1973 | Scrubland | 39 | 0 |
| *Arizmendi, Espinosa & Ornelas (1994)* | Book chapter | 1989–1994 | REPSA | 105 | 5 |
| *Arenas (2004)* | Thesis | September 2001–August 2002 | REPSA | 87 | 0 |
| *Chávez & Gurrola (2007)* | Book chapter | September 2006 | A3 | 66 | 1 |
| *Díaz (2008)* | Thesis | July 2002–September 2003 | JB | 79 | 3 |
| *Chávez & Gurrola (2009)* | Book chapter | Seven years monitoring | REPSA | 148 | 27 |
| *Montiel-Parra et al. (2009)* | Book chapter | Review of specimens collected | REPSA | 10 | 0 |
| *San José, Garmendia & Cano-Santana (2010)* | Scientific article | June 2009–May 2010 | A11, ZNP | 78 | 2 |
| *Aguilar-Gómez, Calderón-Parra & Ortega-Álvarez (2015)* | Scientific article | 25 January 2014 | ZNP | 1 | 1 |
| *Andrade (2016)* | Thesis | August 2014–November 2015 | A3 | 2 | 0 |
| *Gallegos (2016)* | Thesis | September 2012–August 2013 | A3 | 70 | 3 |
| | | | JB | 72 | |
| *Rodríguez (2020)* | Thesis | March–October 2019 | JB | 6 | 0 |
| Citizen science | Databases | 2005–2023 | JB, A3, REPSA | 198 | 49 |
| | | 1967–2023 | JB, A3, A12, ZNP, REPSA | 176 | |
| This study | Biological monitoring | December 2013–December 2023 | JB, ZNP, A11, A12, A13, A5, ZNSO, A3 | 157 | 157 |

pink squares and pink line). The nestedness showed similar values throughout all census intervals (Fig. 2, orange triangles and orange line).

The GLMs indicated that time and species residency status did not have significant effect on the proportion of species that become extinct in the reserve (Table 3, Fig. 3A). The proportion of species arriving in the reserve followed a similar trend and was not influenced by time or residency status of the species (Table 3, Fig. 3B).

Based on the percentage of persistence of each species in the REPSA, we detected 58 core species (*i.e.*, species recorded in more than 66% of the lists evaluated; Fig. 4, green boxes). Among the core species, 35 were resident, 22 have some migratory movement and

**Table 2 Historic checklist of bird species recorded in the REPSA from 1967 to 2023 following the taxonomy proposed by *Chesser et al. (2023)*.** Categories of residency and abundance of each species in the REPSA are indicated, as well as the source of information associated with each record. The percentage of persistence and whether the species is core (C), intermediate (I) or transient (T) is indicated. Residency categories in REPSA: Resident, R; Winter migrant, MI ; Summer migrant, MV; Transient migrant, TM; Escapes, E. Abundance categories: Very abundant, MA; abundant, A; common, C; rare, R; very rare, MR. Sources of information: *Ramos (1974)* = 1; *González (1984)* = 2; *Arizmendi, Espinosa & Ornelas (1994)* = 3; *Chávez & Gurrola (2007)* = 4; *Díaz (2008)* = 5; *Chávez & Gurrola (2009)* = 6; *San José, Garmendia & Cano-Santana (2010)* = 7; *Aguilar-Gómez, Calderón-Parra & Ortega-Álvarez (2015)* = 8; *Gallegos (2016)* = 9; Citizen sciene, iNatualist and eBird = 10 and this study = 11. Species with insufficient information to determine its residency pattern within the reserve where marked with an asterisk (*). For these species, the main category of residence in Mexico is indicated (*Berlanga et al., 2015*).

| Order<br>Family<br>*Species* | Common name | Residency | Abundance | Source | Persistence category | Persistence (%) | Species code |
|---|---|---|---|---|---|---|---|
| **Anseriformes** | | | | | | | |
| Anatidae | | | | | | | |
| *Aix sponsa* | Wood duck | MI* | MR | 11 | T | 7 | Aspo |
| *Spatula discors* | Blue-winged teal | MI* | MR | 6 | T | 13 | Sdis |
| *Spatula clypeata* | Northern shoveler | MI | R | 11 | T | 33 | Scly |
| *Mareca strepera* | Gadwall | MI* | MR | 10 | T | 7 | Mstr |
| *Mareca americana* | American wigeon | MI | MR | 9 | T | 7 | Aame |
| *Anas diazi* | Mexican duck | R | A | 11 | I | 60 | Adia |
| **Podicipediformes** | | | | | | | |
| Podicipedidae | | | | | | | |
| *Tachybaptus dominicus* | Least grebe | R* | R | 11 | T | 13 | Tdom |
| *Podilymbus podiceps* | Pied-billed grebe | R* | R | 11 | T | 33 | Ppod |
| *Podiceps nigricollis* | Eared grebe | MI* | MR | 11 | T | 7 | Pnig |
| **Columbiformes** | | | | | | | |
| Columbidae | | | | | | | |
| *Columba livia* | Rock pigeon | E | C | 11 | I | 53 | Cliv |
| *Streptopelia decaocto* | Eurasian collared-dove | E | MR | 11 | T | 7 | Sdec |
| *Columbina inca* | Inca dove | R | MA | 11 | C | 100 | Cinc |
| *Columbina passerina* | Common ground dove | R* | MR | 11 | T | 7 | Colpass |
| *Zenaida asiatica* | White-winged dove | MI | MR | 11 | T | 20 | Zasi |
| *Zenaida macroura* | Mourning dove | MI | MR | 11 | T | 33 | Zmac |
| **Cuculiformes** | | | | | | | |
| Cuculidae | | | | | | | |
| *Crotophaga sulcirostris* | Groove-billed Ani | R* | MR | 11 | T | 7 | Csul |
| *Coccyzus americanus* | Yellow-billed cuckoo | MV* | MR | 6 | T | 20 | Cocame |
| **Caprimulgiformes** | | | | | | | |
| Caprimulgidae | | | | | | | |
| *Chordeiles acutipennis* | Lesser nighthawk | MV* | MR | 11 | T | 13 | Cacu |
| *Antrostomus arizonae* | Mexican whip-poor-will | R* | MR | 6 | T | 13 | Aari |
| **Apodiformes** | | | | | | | |
| Apodidae | | | | | | | |
| *Cypseloides niger* | Black swift | MV | C | 11 | I | 47 | Cnig |
| *Streptoprocne rutila* | Chestnut-collared swift | R* | MR | 10 | T | 27 | Streru |
| *Streptoprocne semicollaris* | White-naped swift | R* | R | 11 | T | 13 | Ssem |

| Order Family *Species* | Common name | Residency | Abundance | Source | Persistence category | Persistence (%) | Species code |
|---|---|---|---|---|---|---|---|
| *Chaetura vauxi* | Vaux's swift | R | C | 11 | I | 47 | Cvau |
| *Aeronautes saxatalis* | White-throated swift | R* | MR | 10 | T | 7 | Asax |
| Trochilidae | | | | | | | |
| *Colibri thalassinus* | Mexican violetear | R* | MR | 10 | T | 13 | Ctha |
| *Eugenes fulgens* | Rivoli's hummingbird | R | C | 11 | C | 100 | Euful |
| *Heliomaster constantii* | Plain-capped starthroat | MI | MR | 10 | T | 7 | Hcon |
| *Lampornis clemenciae* | Blue-throated mountain-gem | R* | MR | 11 | I | 47 | Lcle |
| *Tilmatura dupontii* | Sparkling-tailed hummingbird | R* | MR | 10 | T | 7 | Tdup |
| *Calothorax lucifer* | Lucifer hummingbird | R | R | 11 | I | 60 | Cluc |
| *Archilochus colubris* | Ruby-throated hummingbird | MI | R | 11 | T | 27 | Acol |
| *Archilochus alexandri* | Black-chinned hummingbird | MI | MR | 11 | T | 7 | Aale |
| *Selasphorus calliope* | Calliope hummingbird | MI* | MR | 6 | T | 13 | Scal |
| *Selasphorus rufus* | Rufous hummingbird | MI | MR | 10 | T | 27 | Sruf |
| *Selasphorus sasin* | Allen's hummingbird | MI* | MR | 3 | T | 7 | Ssas |
| *Selasphorus platycercus* | Broad-tailed hummingbird | MI | MR | 11 | T | 27 | Spla |
| Selasphorus heloisa | Bumblebee hummingbird | R* | MR | 11 | T | 13 | Shel |
| *Cynanthus latirostris* | Broad-billed hummingbird | R | MA | 11 | C | 100 | Clat |
| *Basilinna leucotis* | White-eared hummingbird | R* | MR | 11 | I | 60 | Bleu |
| Ramosomyia violiceps | Violet-crowned hummingbird | MI | R | 11 | I | 53 | Rvio |
| Saucerottia beryllina | Berylline hummingbird | R | MA | 11 | C | 100 | Sber |
| **Gruiformes** | | | | | | | |
| Rallidae | | | | | | | |
| *Gallinula galeata* | Common gallinule | R | C | 11 | I | 47 | Ggal |
| *Fulica americana* | American coot | R | C | 11 | I | 47 | Fame |
| **Charadriiformes** | | | | | | | |
| Charadriidae | | | | | | | |
| *Charadrius vociferus* | Killdeer | R* | MR | 10 | T | 13 | Cvoc |
| *Charadrius semipalmatus* | Semipalmated plover | MI* | MR | 6 | T | 7 | Csem |
| Scolopacidae | | | | | | | |
| *Actitis macularius* | Spotted sandpiper | MI* | MR | 6 | T | 7 | Amac |
| **Pelecaniformes** | | | | | | | |
| Pelecanidae | | | | | | | |
| *Pelecanus erythrorhynchos* | American white pelican | MI | C | 11 | T | 7 | Pery |
| Ardeidae | | | | | | | |
| *Ardea herodias* | Great blue heron | MI | R | 11 | I | 47 | Aher |
| *Ardea alba* | Great egret | MI | C | 11 | T | 27 | Aalb |
| *Egretta tricolor* | Tricolored heron | MI* | R | 11 | T | 7 | Etri |
| *Butorides virescens* | Green heron | R | C | 11 | C | 67 | Bvires |
| *Nycticorax nycticorax* | Black-crowned night-heron | R | C | 11 | I | 53 | Nnyc |

*(Continued)*

| Order<br>Family<br>*Species* | Common name | Residency | Abundance | Source | Persistence<br>category | Persistence<br>(%) | Species<br>code |
|---|---|---|---|---|---|---|---|
| Threskiornithidae | | | | | | | |
| *Plegadis chihi* | White-faced Ibis | MI* | MR | 10 | T | 7 | Pchi |
| **Cathartiformes** | | | | | | | |
| Cathartidae | | | | | | | |
| *Cathartes aura* | Turkey vulture | R* | R | 11 | T | 13 | Caura |
| **Accipitriformes** | | | | | | | |
| Pandionidae | | | | | | | |
| *Pandion haliaetus* | Osprey | MI* | MR | 10 | T | 7 | Phal |
| Accipitridae | | | | | | | |
| *Chondrohierax uncinatus* | Hook-billed kite | MI | C | 11 | I | 53 | Cunc |
| *Circus hudsonius* | Northern harrier | MI* | MR | 6 | T | 20 | Chud |
| *Accipiter striatus* | Sharp-shinned hawk | R | R | 11 | I | 47 | Astr |
| *Accipiter cooperii* | Cooper's 'awk | R | C | 11 | C | 67 | Acoo |
| *Buteogallus anthracinus* | Common black hawk | R* | MR | 5 | T | 7 | Bant |
| *Parabuteo unicinctus* | Harris's hawk | R | R | 11 | T | 27 | Puni |
| *Buteo lineatus* | Red-shouldered hawk | MI | MR | 11 | T | 7 | Blin |
| *Buteo platypterus* | Broad-winged hawk | TM* | R | 11 | T | 13 | Bpla |
| *Buteo brachyurus* | Short-tailed hawk | R* | MR | 11 | T | 7 | Bubrac |
| *Buteo swainsoni* | Swainson's hawk | TM* | MR | 3 | T | 7 | Bswa |
| *Buteo jamaicensis* | Red-tailed hawk | R | R | 11 | T | 33 | Bjam |
| **Strigiformes** | | | | | | | |
| Tytonidae | | | | | | | |
| *Tyto alba* | Barn owl | R* | MR | 11 | T | 7 | Talb |
| Strigidae | | | | | | | |
| *Megascops kennicottii* | Western screech-owl | R* | MR | 10 | T | 13 | Mken |
| *Megascops asio* | Eastern screech-owl | R* | MR | 1 | T | 7 | Masi |
| *Bubo virginianus* | Great horned owl | R | R | 11 | T | 33 | Bvirgi |
| **Trogoniformes** | | | | | | | |
| Trogonidae | | | | | | | |
| *Trogon elegans* | Elegant trogon | R* | MR | 11 | T | 7 | Tele |
| **Coraciiformes** | | | | | | | |
| Alcedinidae | | | | | | | |
| *Megaceryle alcyon* | Belted kingfisher | MI | MR | 11 | T | 13 | Malc |
| **Piciformes** | | | | | | | |
| Picidae | | | | | | | |
| *Melanerpes formicivorus* | Acorn woodpecker | R* | MR | 10 | T | 7 | Mfor |
| *Melanerpes chrysogenys* | Golden-cheeked woodpecker | R* | MR | 2 | T | 7 | Mchr |
| *Melanerpes aurifrons* | Golden-fronted woodpecker | R* | MR | 11 | T | 7 | Melauri |
| *Sphyrapicus varius* | Yellow-bellied sapsucker | MI | R | 11 | T | 27 | Svar |
| *Dryobates scalaris* | Ladder-backed woodpecker | R | A | 11 | C | 100 | Dsca |

| Order Family *Species* | Common name | Residency | Abundance | Source | Persistence category | Persistence (%) | Species code |
|---|---|---|---|---|---|---|---|
| *Dryobates villosus* | Hairy woodpecker | R* | MR | 10 | T | 7 | Dvil |
| *Colaptes auratus* | Northen flicker | R* | MR | 6 | T | 13 | Colaur |
| **Falconiformes** | | | | | | | |
| Falconidae | | | | | | | |
| *Falco sparverius* | American kestrel | MI | C | 11 | C | 80 | Fspa |
| *Falco columbarius* | Merlin | MI | MR | 6 | T | 7 | Fcol |
| *Falco peregrinus* | Peregrine falcon | MI | MR | 6 | T | 7 | Fper |
| *Falco mexicanus* | Praire falcon | MI | MR | 6 | T | 7 | Fmex |
| **Psittaciformes** | | | | | | | |
| Psittacidae | | | | | | | |
| *Myiopsitta monachus* | Monk parakeet | E | MR | 11 | T | 7 | Mmon |
| *Eupsittula canicularis* | Orange-fronted parakeet | E | MR | 6 | T | 7 | Ecan |
| *Amazona albifrons* | White.fronted parrot | E | MR | 6 | T | 13 | Aalbi |
| *Amazona viridigenalis* | Red-crowned parrot | E | MR | 6 | T | 7 | Avir |
| *Amazona autumnalis* | Red-lored parrot | E | MR | 6 | T | 20 | Aaut |
| *Amazona oratrix* | Yellow-headed parrot | E | MR | 6 | T | 7 | Aora |
| Psittaculidae | | | | | | | |
| *Agapornis personatus* | Yellow-collared lovebird | E | MR | 11 | T | 7 | Aper |
| *Agapornis roseicollis* | Rosy-faced lovebird | E | MR | 10 | T | 7 | Aros |
| **Passeriformes** | | | | | | | |
| Tyrannidae | | | | | | | |
| *Camptostoma imberbe* | Northern beardless-tyrannulet | R | R | 11 | I | 53 | Cimb |
| *Myiopagis viridicata* | Greenish elaenia | R* | MR | 11 | T | 13 | Mvir |
| *Myiarchus tuberculifer* | Dusky-capped flycathcer | R* | MR | 10 | T | 27 | Mtub |
| *Myiarchus cinerascens* | Ash-throated flycatcher | MI | R | 11 | I | 47 | Mcin |
| *Pitangus sulphuratus* | Great kiskadee | R* | MR | 10 | T | 7 | Psul |
| *Tyrannus melancholicus* | Tropical kingbird | R* | MR | 5 | T | 7 | Tmel |
| *Tyrannus vociferans* | Cassin's kingbird | R | MA | 11 | C | 93 | Tvoc |
| *Tyrannus verticalis* | Western kingbird | MI* | MR | 11 | T | 13 | Tver |
| *Tyrannus tyrannus* | Eastern kingbird | TM* | R | 11 | T | 7 | Ttyr |
| *Tyrannus forficatus* | Scissor-tailed flycatcher | MI* | MR | 11 | T | 20 | Tfor |
| *Mitrephanes phaeocercus* | Tufted flycatcher | R* | MR | 10 | T | 7 | Mpha |
| *Contopus cooperi* | Olive-sided flycatcher | MI | R | 11 | T | 20 | Ccoo |
| *Contopus pertinax* | Greater pewee | R | C | 11 | C | 93 | Cper |
| *Contopus sordidulus* | Western wood-pewee | MI | R | 11 | I | 60 | Csor |
| *Contopus virens* | Eastern wood-pewee | TM* | MR | 2 | T | 7 | Cvir |
| *Empidonax albigularis* | White-throated flycatcher | MI* | MR | 10 | T | 7 | Ealb |
| *Empidonax minimus* | Least flycatcher | MI | R | 11 | I | 53 | Emin |
| *Empidonax hammondii* | Hammond's flycatcher | MI | C | 11 | I | 60 | Eham |
| *Empidonax wrightii* | Gray flycatcher | MI* | MR | 2 | T | 7 | Ewri |

*(Continued)*

| Order Family *Species* | Common name | Residency | Abundance | Source | Persistence category | Persistence (%) | Species code |
|---|---|---|---|---|---|---|---|
| *Empidonax oberholseri* | Dusky flycatcher | MI | C | 11 | I | 60 | Eobe |
| *Empidonax difficilis* | Western flycatcher | R | C | 11 | I | 60 | Edif |
| *Empidonax fulvifrons* | Buff-breasted flycatcher | R | A | 11 | C | 80 | Emful |
| *Sayornis nigricans* | Balck phoebe | R* | MR | 11 | T | 20 | Saynig |
| *Sayornis phoebe* | Eastern phoebe | MI* | MR | 10 | T | 7 | Spho |
| *Sayornis saya* | Say's phoebe | R* | MR | 2 | T | 7 | Ssay |
| *Pyrocephalus rubinus* | Vermilion flycatcher | R | MA | 11 | C | 93 | Pyrub |
| Vireonidae | | | | | | | |
| *Vireo atricapilla* | Black-capped vireo | MI* | MR | 4 | T | 7 | Vatr |
| *Vireo nelsoni* | Dwarf vireo | R* | MR | 10 | T | 7 | Vnel |
| *Vireo bellii* | Bell's vireo | MI* | MR | 10 | T | 27 | Vbel |
| *Vireo huttoni* | Hutton's vireo | MI | R | 11 | I | 60 | Vhut |
| *Vireo cassinii* | Cassin's vireo | MI | MR | 11 | T | 13 | Vcas |
| *Vireo solitarius* | Blue-headed vireo | MI* | MR | 5 | T | 20 | Vsol |
| *Vireo plumbeus* | Plumbeous vireo | MI* | R | 11 | T | 20 | Vplu |
| *Vireo philadelphicus* | Philadelphia vireo | MI* | MR | 8 | T | 7 | Vphi |
| *Vireo gilvus* | Warbling vireo | MI | C | 11 | C | 67 | Vgil |
| Laniidae | | | | | | | |
| *Lanius ludovicianus* | Loggerhead shrike | MI | R | 11 | I | 60 | Llud |
| Corvidae | | | | | | | |
| *Calocitta colliei* | Black-throated magpie-jay | R* | MR | 9 | T | 7 | Ccol |
| *Calocitta formosa* | White-throated magpie-jay | E | MR | 6 | T | 20 | Cfor |
| *Psilorhinus morio* | Brown jay | E | MR | 11 | T | 7 | Pmor |
| *Cyanocorax yncas* | Green jay | R | R | 11 | I | 47 | Cync |
| *Cyanocorax sanblasianus* | San Blas jay | E | MR | 6 | T | 7 | Csan |
| *Cyanocitta stelleri* | Steller's jay | R* | MR | 6 | T | 20 | Cste |
| *Aphelocoma woodhouseii* | Woodhouse's scrub-jay | R | C | 11 | C | 67 | Awoo |
| *Aphelocoma ultramarina* | Transvolcanic Jay | R | MR | 10 | T | 20 | Ault |
| *Corvus corax* | Common raven | R* | MR | 6 | T | 7 | Ccor |
| Paridae | | | | | | | |
| *Poecile sclateri* | Mexican chickadee | R* | MR | 10 | T | 13 | Pscl |
| Hirundinidae | | | | | | | |
| *Riparia riparia* | Bank swallow | TM* | MR | 10 | T | 7 | Rrip |
| *Tachycineta bicolor* | Tree swallow | MI* | MR | 7 | T | 7 | Tbic |
| *Tachycineta thalassina* | Violet-green swallow | R* | MR | 10 | T | 7 | Ttha |
| *Stelgidopteryx serripennis* | Northern rough-winged swallow | MI | C | 11 | I | 60 | Sser |
| *Hirundo rustica* | Barn swallow | MV | MA | 11 | C | 100 | Hrus |
| *Petrochelidon pyrrhonota* | Cliff swallow | MV | A | 11 | C | 73 | Ppyr |

| Order Family Species | Common name | Residency | Abundance | Source | Persistence category | Persistence (%) | Species code |
|---|---|---|---|---|---|---|---|
| Aegithalidae | | | | | | | |
| *Psaltriparus minimus* | Bushtit | R | MA | 11 | C | 100 | Pmin |
| Regulidae | | | | | | | |
| Corthylio calendula | Ruby-crowned kinglet | MI | A | 11 | C | 100 | Ccal |
| *Regulus satrapa* | Golden-crowned kinglet | R* | MR | 10 | T | 7 | Rsat |
| Bombycillidae | | | | | | | |
| *Bombycilla cedrorum* | Cedar waxwing | MI | C | 11 | I | 47 | Bced |
| Ptiliogonatidae | | | | | | | |
| *Ptiliogonys cinereus* | Gray silky-flycatcher | R | MA | 11 | C | 80 | Pcin |
| Sittidae | | | | | | | |
| *Sitta carolinensis* | White-breasted nuthatch | MI | MR | 11 | T | 13 | Scar |
| *Sitta pygmaea* | Pygmy nuthatch | R* | MR | 10 | T | 7 | Spyg |
| Certhiidae | | | | | | | |
| *Certhia americana* | Brown creeper | R* | MR | 10 | T | 7 | Cerame |
| Polioptilidae | | | | | | | |
| *Polioptila caerulea* | Blue-gray gnatcatcher | MI | A | 11 | C | 100 | Polcae |
| Troglodytidae | | | | | | | |
| *Salpinctes obsoletus* | Rock wren | R* | MR | 1 | T | 13 | Sobs |
| *Catherpes mexicanus* | Canyon wren | R | A | 11 | C | 100 | Cmex |
| *Campylorhynchus megalopterus* | Gray-barred wren | R* | MR | 10 | T | 7 | Cmeg |
| *Thryomanes bewickii* | Bewick's wren | R | MA | 11 | C | 93 | Tbew |
| *Troglodytes aedon* | House wren | R | C | 11 | C | 87 | Taed |
| *Cistothorus palustris* | Marsh wren | MI | MR | 11 | T | 13 | Cpal |
| Mimidae | | | | | | | |
| *Melanotis caerulescens* | Blue mockingbird | R | C | 11 | C | 67 | Mcae |
| *Dumetella carolinensis* | Gray catbird | MI | MR | 11 | T | 7 | Dcar |
| *Toxostoma curvirostre* | Curve-billed thrasher | R | A | 11 | C | 100 | Tcur |
| *Toxostoma ocellatum* | Ocellated thrasher | R* | MR | 1 | T | 7 | Toce |
| *Mimus polyglottos* | Northern mockingbird | MI | R | 11 | I | 47 | Mpol |
| Sturnidae | | | | | | | |
| *Sturnus vulgaris* | European starling | R | MR | 6 | T | 13 | Svul |
| Turdidae | | | | | | | |
| *Myadestes occidentalis* | Brown-backed solitaire | R* | MR | 10 | T | 7 | Mocc |
| *Catharus aurantiirostris* | Orange-billed nightingale-thrush | R* | MR | 1 | T | 7 | Caur |
| *Catharus occidentalis* | Russet nightingale-thrush | R* | MR | 11 | T | 7 | Cocc |
| *Catharus ustulatus* | Swainson's thrush | TM* | MR | 6 | T | 27 | Cust |
| *Catharus guttatus* | Hermit thrush | MI* | R | 11 | I | 47 | Cgut |
| *Turdus grayi* | Clay-colored thrush | R* | MR | 2 | T | 7 | Tgra |

(Continued)

| Order Family Species | Common name | Residency | Abundance | Source | Persistence category | Persistence (%) | Species code |
|---|---|---|---|---|---|---|---|
| *Turdus assimilis* | White-throated thrush | R* | MR | 10 | T | 7 | Tass |
| *Turdus rufopalliatus* | Rufous-backed robin | R | A | 11 | C | 100 | Truf |
| *Turdus migratorius* | American robin | R | MA | 11 | C | 100 | Tmig |
| Peucedramidae | | | | | | | |
| *Peucedramus taeniatus* | Olive warbler | R | R | 11 | T | 20 | Ptae |
| Passeridae | | | | | | | |
| *Passer domesticus* | House sparrow | E | MA | 11 | C | 87 | Pdom |
| Fringillidae | | | | | | | |
| *Chlorophonia elegantissima* | Elegant euphonia | R | R | 11 | T | 13 | Cele |
| *Haemorhous mexicanus* | House finch | R | MA | 11 | C | 100 | Hmex |
| *Loxia curvirostra* | Red crossbill | R* | MR | 3 | T | 13 | Lcur |
| *Spinus pinus* | Pine siskin | R* | MR | 10 | T | 7 | Spin |
| *Spinus notatus* | Black-headed siskin | R* | MR | 3 | T | 7 | Snot |
| *Spinus psaltria* | Lesser goldfinch | R | MA | 11 | C | 93 | Spsa |
| Passerellidae | | | | | | | |
| *Ammodramus savannarum* | Grasshopper sparrow | MI* | MR | 10 | T | 7 | Asav |
| *Chondestes grammacus* | Lark sparrow | MI | C | 11 | I | 47 | Cgra |
| *Spizella passerina* | Chipping sparrow | R | A | 11 | C | 100 | Spas |
| *Spizella pallida* | Clay-colored sparrow | MI | C | 11 | I | 47 | Spal |
| *Spizella atrogularis* | Black-chinned sparrow | R | C | 11 | C | 87 | Satr |
| *Arremon virenticeps* | Green-striped brush-finch | R* | MR | 10 | T | 13 | Avire |
| *Junco phaeonotus* | Yellow-eyed junco | R* | MR | 6 | T | 13 | Jpha |
| *Oriturus superciliosus* | Striped sparrow | R* | MR | 6 | T | 7 | Osupe |
| *Pooecetes gramineus* | Vesper sparrow | MI* | R | 11 | T | 27 | Pgra |
| *Passerculus sandwichensis* | Savannah sparrow | MI* | MR | 10 | T | 13 | Psan |
| *Melospiza melodia* | Song sparrow | R | A | 11 | C | 80 | Mmel |
| *Melospiza lincolnii* | Lincoln's sparrow | MI | C | 11 | C | 80 | Mlin |
| *Melozone kieneri* | Rusty-crowned ground-sparrow | R | R | 11 | T | 27 | Mkie |
| *Melozone fusca* | Canyon towhee | R | MA | 11 | C | 100 | Mfus |
| *Aimophila ruficeps* | Rufous-crowned sparrow | R | A | 11 | C | 93 | Aruf |
| *Pipilo chlorurus* | Green-tailed towhee | MI* | MR | 10 | T | 7 | Pchl |
| *Pipilo maculatus* | Spotted towhee | R* | MR | 3 | T | 7 | Pmac |
| *Atlapates pileatus* | Rufous-capped brushfinch | R* | MR | 6 | T | 20 | Apil |
| Icteriidae | | | | | | | |
| *Icteria virens* | Yellow-breasted chat | MI | MR | 11 | T | 20 | Ivir |
| Icteridae | | | | | | | |
| *Icterus wagleri* | Black-vented oriole | R | MR | 11 | T | 7 | Iwag |
| *Icterus spurius* | Orchad oriole | MI | C | 11 | C | 67 | Ispu |
| *Icterus cucullatus* | Hooded oriole | MI* | R | 11 | T | 33 | Icuc |

| Order<br>Family<br>*Species* | Common name | Residency | Abundance | Source | Persistence category | Persistence (%) | Species code |
|---|---|---|---|---|---|---|---|
| *Icterus bullockii* | Bullock's oriole | MI | C | 11 | C | 73 | Ibul |
| *Icterus gularis* | Altamira oriole | E | MR | 10 | T | 20 | Igul |
| *Icterus gálbula* | Baltimore oriole | MI* | MR | 7 | T | 27 | Igal |
| *Icterus abeillei* | Black-backed oriole | R | A | 11 | C | 87 | Iabe |
| *Icterus parisorum* | Scott's oriole | R | C | 11 | C | 100 | Ipar |
| *Agelaius phoeniceus* | Red-winged blackbird | R | MR | 11 | T | 20 | Apho |
| *Molothrus aeneus* | Bronzed cowbird | MV | A | 11 | C | 93 | Maen |
| *Molothrus ater* | Brown-headed cowbird | MI | R | 11 | T | 27 | Mate |
| *Dives dives* | Melodious blackbird | R* | MR | 2 | T | 7 | Ddiv |
| *Quiscalus mexicanus* | Great-tailed grackle | R | C | 11 | I | 53 | Qmex |
| Parulidae | | | | | | | |
| *Seiurus aurocapilla* | Ovenbird | MI | MR | 10 | T | 13 | Saur |
| *Parkesia motacilla* | Louisiana waterthrush | MI | R | 11 | T | 13 | Pmot |
| *Parkesia noveboracensis* | Northern waterthrush | MI | R | 11 | I | 53 | Pnov |
| *Mniotilta varia* | Black-and-white warbler | MI | C | 11 | C | 93 | Mvar |
| *Protonotaria citrea* | Prothonotary warbler | MI* | MR | 10 | T | 7 | Pcit |
| *Oreothlypis superciliosa* | Crescent-chested warbler | R* | MR | 11 | T | 7 | Osup |
| *Leiothlypis peregrina* | Tennesse warbler | MI* | MR | 6 | T | 20 | Lper |
| *Leiothlypis celata* | Orange-crowned warbler | MI | A | 11 | C | 100 | Lcel |
| *Leiothlypis ruficapilla* | Nashville warbler | MI | A | 11 | C | 93 | Lruf |
| *Leiothlypis virginiae* | Virginia's warbler | MI* | MR | 10 | T | 27 | Lvir |
| *Geothlypis tolmiei* | MacGillivray's warbler | MI | C | 11 | C | 93 | Gtol |
| *Geothlypis trichas* | Common yellowthroat | MI | R | 11 | T | 33 | Gtri |
| *Geothlypis nelsoni* | Hooded yellowthroat | R | A | 11 | C | 87 | Gnel |
| Setophaga citrina | Hooded warbler | MI* | MR | 10 | T | 7 | Scit |
| *Setophaga ruticilla* | American redstart | MI* | MR | 11 | T | 20 | Serut |
| *Setophaga americana* | Northern Parula | MI* | MR | 10 | T | 13 | Setame |
| *Setophaga petechia* | Yellow warbler | MI* | MR | 10 | T | 7 | Spete |
| *Setophaga coronata* | Yellow-rumped warbler | MI | MA | 11 | C | 100 | Scor |
| *Setophaga dominica* | Yellow-thoratet warbler | MI* | MR | 10 | T | 7 | Sdom |
| *Setophaga graciae* | Grace's warbler | R* | MR | 10 | T | 7 | Sgra |
| *Setophaga nigrescens* | Black-throated gray warbler | MI | C | 11 | C | 73 | Setnig |
| *Setophaga townsendi* | Townsend's warbler | MI | C | 11 | C | 80 | Stow |
| *Setophaga occidentalis* | Hermit warbler | MI | C | 11 | C | 73 | Socc |
| *Setophaga virens* | Black-throated green warbler | MI | R | 11 | T | 20 | Svir |
| *Basileuterus rufifrons* | Rufous-capped warbler | R* | MR | 6 | T | 33 | Bruf |
| *Cardellina pusilla* | Wilson's warbler | MI | A | 11 | C | 93 | Cpus |
| *Cardellina rubra* | Red warbler | R* | MR | 10 | T | 7 | Crub |
| *Myioborus pictus* | Painted redstart | R* | MR | 10 | T | 7 | Mpic |
| *Myioborus miniatus* | Slate-throated redstart | MI | C | 11 | C | 87 | Mmin |

(Continued)

| Order Family *Species* | Common name | Residency | Abundance | Source | Persistence category | Persistence (%) | Species code |
|---|---|---|---|---|---|---|---|
| Cardinalidae | | | | | | | |
| *Piranga flava* | Hepatic tananger | R* | MR | 11 | T | 33 | Pfla |
| *Piranga rubra* | Summer tananger | MI | C | 11 | C | 87 | Pirub |
| *Piranga ludoviciana* | Western tananger | MI | C | 11 | C | 87 | Pirlud |
| *Piranga bidentata* | Flame-colored tananger | R* | MR | 9 | T | 7 | Pbid |
| *Cardinalis cardinalis* | Northern cardinal | R | A | 11 | C | 87 | Ccar |
| *Pheucticus ludovicianus* | Rose-breasted grosbeak | R* | R | 11 | T | 27 | Phelud |
| *Pheucticus melanocephalus* | Black-headed grosbeak | R | A | 11 | C | 100 | Pmel |
| *Passerina caerulea* | Blue grosbeak | R | A | 11 | C | 87 | Pascae |
| *Passerina amoena* | Lazuli bunting | MI | MR | 10 | T | 7 | Pamo |
| *Passerina cyanea* | Indigo bunting | MI | R | 11 | T | 33 | Pcya |
| *Passerina versicolor* | Varied bunting | MV | R | 11 | I | 40 | Pver |
| *Passerina ciris* | Painted bunting | MI | MR | 11 | T | 27 | Pcir |
| *Spiza americana* | Dickcissel | TM* | R | 11 | T | 7 | Same |
| Thraupidae | | | | | | | |
| *Sicalis luteola* | Grassland yellow-finch | R* | MR | 10 | T | 7 | Slut |
| *Diglossa baritula* | Cinnamon-bellied flowerpiercer | R | A | 11 | C | 93 | Dbar |
| *Sporophila torqueola* | Cinnamon-rumped seedeater | MV | C | 11 | T | 27 | Stor |

one exotic species (Fig. 4, Table 2). Additionally, we detected that 20 of the core species had 100% persistence in the REPSA (Table 2); that is, their presence has been recorded in every listing from the reserve from 1973 to 2023. Of these, 15 species were residents and five had some migratory movement (Table 2). We detected that 33 species had intermediate persistence (33–66% persistence over time, Fig. 4, purple boxes). Of these species, 13 are residents, 19 have some migratory movement and one exotic species. While 167 species were classified as transient since their persistence in the REPSA was less than 33%.

The cumulative records of species with 100% persistence (20 species, 6,406 records) represented 58% of the total bird records. On the other hand, the sum of the records of all core species (58 species, 9,885 records) represented 90% of the total records of avifauna in the REPSA (Supplemental Information).

## DISCUSSION

The historical checklist of avifauna reported in the REPSA comprised 258 species, reported from 1967 to 2023. In Mexico, the presence of between 1,123–1,150 bird species is reported (*Navarro-Sigüenza et al., 2014*), while 355 species occur in Mexico City (*Meléndez-Herrada, Gómez de Silva & Ortega-Álvarez, 2016*). Thus, 22% of the country's bird species richness and 72% of the bird species reported for Mexico City have been recorded in the

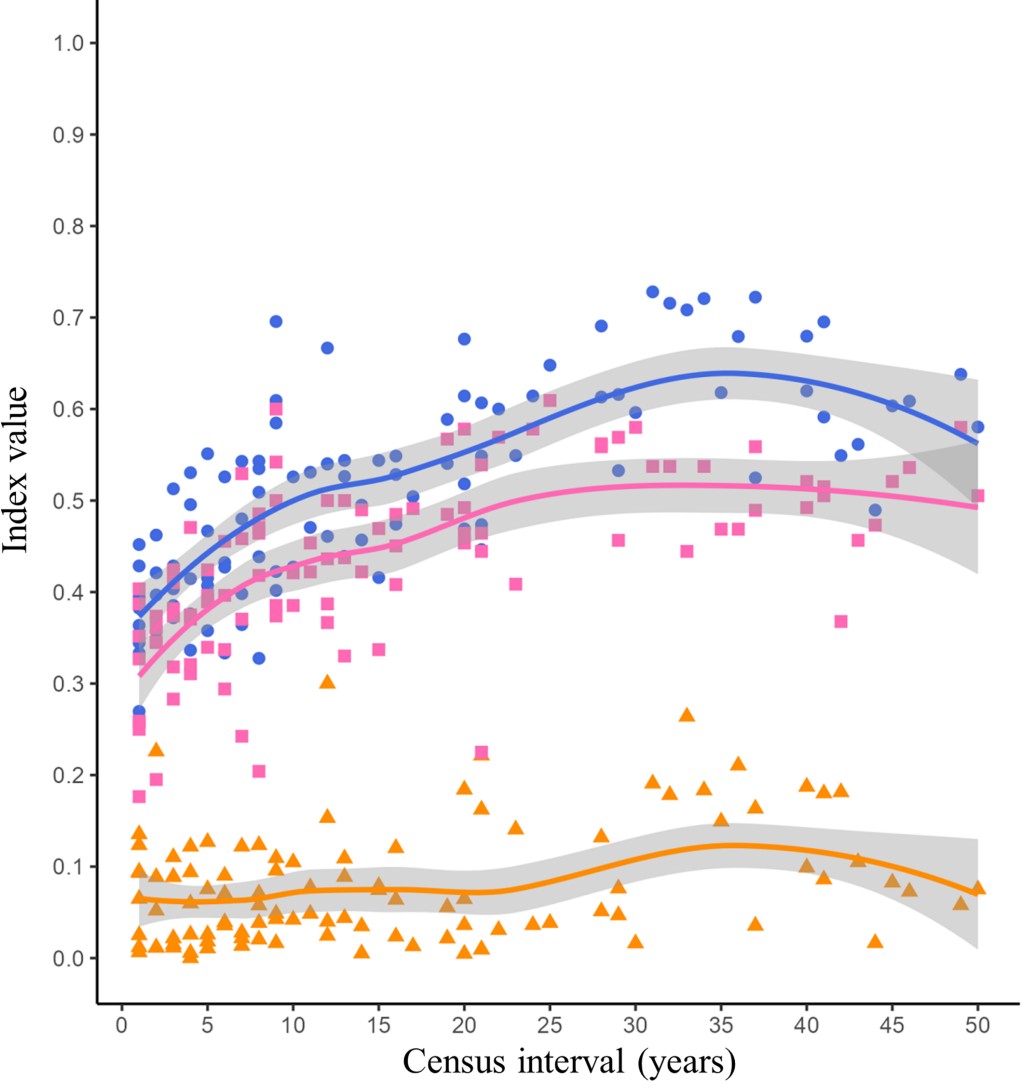

**Figure 2 Jaccard's dissimilarity index and its components (turnover and nestedness) as function of census interval.** Figures and colors correspond to Jaccard's index (blue dots), turnover (pink squares) and nestedness (orange triangles) between annual checklists performed in a given time interval. Lines are the mean values for each variable and gray ribbons are the 95% confidence intervals.

REPSA. The high richness of bird species that have been recorded in the REPSA can be explained by the unique environmental heterogeneity (conditions and resources) that the area presents. It has been reported that vegetation composition and structure are the main factors that determine bird species richness at macrogeographic, regional, and local scales (*Cueto & López de Casenave, 1999*). Areas with a complex vegetation structure and great variety of environmental conditions, as well high plant richness generates a high availability of resources, refuges and nesting sites which promotes high bird richness (*Tu, Fan & Ko, 2020*). Because of its volcanic origin, seven microenvironments have been described within the reserve, each of which presents different conditions in terms of

**Table 3 GLMs summary.** The GLMs evaluates whether census interval between annual checklists, the bird residency status (resident or migratory) and the interactions of these factors influence the bird extinction and bird arrival in the reserve.

| Response variable | Explanatory variables | Estimate | ES | z | *P* |
|---|---|---|---|---|---|
| Bird extinction | Intercept | 0.3617 | 0.3048 | 1.187 | 0.2353 |
| | Census interval | −0.0039 | 0.0145 | −0.273 | 0.7847 |
| | Residency status | −0.7235 | 0.431 | −1.678 | 0.0933 |
| | Census interval * Residency status | 0.0079 | 0.0205 | 0.386 | 0.6993 |
| Bird arrival | Intercept | 0.2415 | 0.3022 | 0.799 | 0.424 |
| | Census interval | −0.0061 | 0.0144 | −0.429 | 0.668 |
| | Residency status | −0.4831 | 0.4274 | −1.13 | 0.258 |
| | Census interval * Residency status | 0.0123 | 0.0203 | 0.606 | 0.544 |

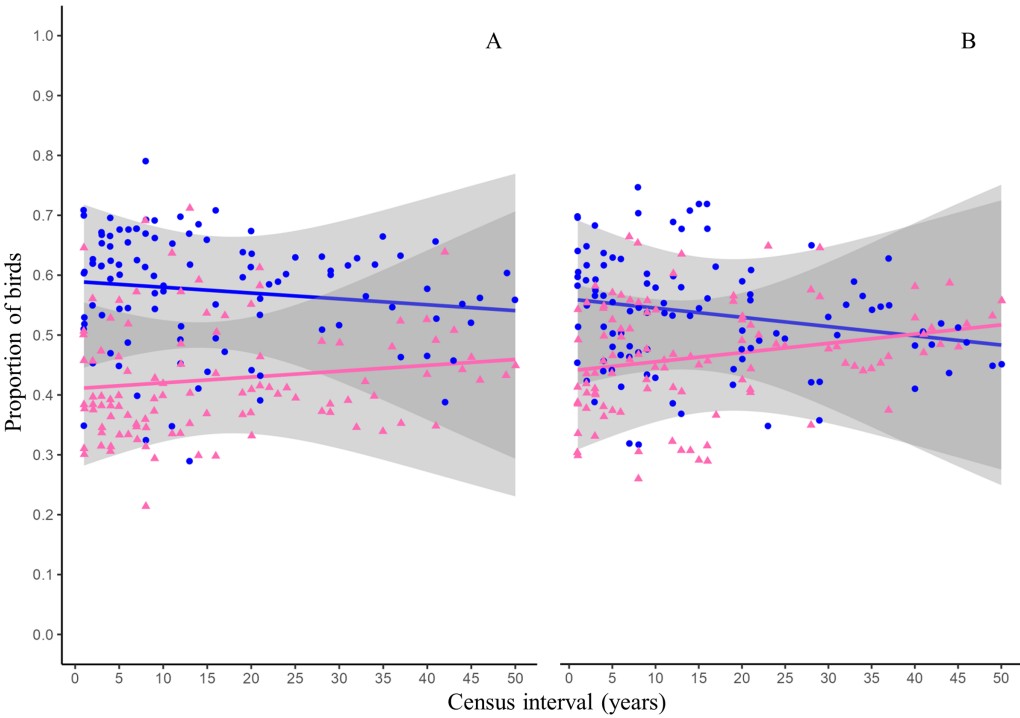

**Figure 3 Proportion of birds becoming extinct and arriving in the reserve as function of census interval.** Figures and colors correspond to proportion of resident (blue dots) and migratory (pink triangles) birds becoming extinct (A) and that arriving in the reserve (B) between annual checklists performed in a given time interval. Lines are the probability that the birds becoming extinct (A) or arriving in the reserve (B) and gray ribbons are the 95% confidence intervals.

humidity, soil depth, and solar incidence, which in turn have allowed the establishment of 377 plant species (*Castillo-Argüero et al., 2009*).

The dissimilarity between the listings of avifauna showed a tendency to increase with increasing census interval (Fig. 2). The greatest source of dissimilarity between annual listings was attributed to species turnover (Fig. 2). As expected, turnover increase with

**Figure 4 Treemap of birds in the REPSA.** Black lines group birds by persistence category. Green boxes indicate core species (C), purple boxes intermediate species (I) and blue boxes transient species (T). Red outlines group species based on their residence in the REPSA: R, resident, MI, winter migrants, MV, summer migrants and E, escapes. The size of the white-bordered squares is proportional to the percentage persistence of each bird species in the REPSA. The acronym for each bird identifier (white text) is the same as in Table 2.

time. This increase was non-linear and asymptotic. *Russell et al. (1995)* postulated that the asymptotic trend in turnover will occur at census intervals between 10 and 50 years. Time range in which asymptotic turnover is reached will depend on intrinsic factors such as the probability of extinction or species arrival in the community and extrinsic factors such as habitat succession (*Russell et al., 1995*). Our results showed that the asymptotic trend in species turnover begins at the 20-year census interval reaching values of 0.50 (Fig. 2, pink line). Long-term research has reported bird species turnover rates similar to what we found in our study. *Diamond (1969)* reported that in the Channel Islands of Southern California the number of bird species remained stable over time (51 years), however species turnover varied among each island from 10 to 62 per cent. Higher species turnover was inversely correlated with total bird richness as well as with plant richness and complexity on each island (*Diamond, 1969*). Highly variable turnover values have been reported in different bird communities, for example in an altitudinal gradient in the Sierra Nevada of southern Spain a species turnover rate of 0.27 evaluated over a 30-year period was reported (*Zamora & Barea-Azcón, 2015*). However, the turnover rate was different in each vegetation type along this altitudinal gradient: high mountain pasture (0.16), Juniperus forest (0.25) and Oak forest (0.35; *Zamora & Barea-Azcón, 2015*). Similarly, in a Mediterranean shrubland in southwestern Spain the average annual bird species turnover rate was 0.24 over a 40-year time scale (*Campo-Celada et al., 2022*). It has been reported that habitats with a more stable vegetation structure over time, such as mature forests, will have a more predictable and stable bird community (*Bengtsson, Baillie & Lawton, 1997*). Due to

succession processes, shrublands could present changes in their vegetation structure over time, increasing the variability in the composition of their bird species. However, specific long-term research is needed to evaluate changes in vegetation structure within the reserve and their effect on bird community composition.

Nestedness was the component that contributed least to the dissimilarity between bird listings. As we expected the number of species recorded was similar over time (Fig. 2, orange line). *MacArthur & Wilson (1963)* reported the positive relationship of number of birds and island area. Larger islands may have greater diversity of habitats that provide a greater gradient of conditions and resources for species (*Lack, 1969*). The reserve was decreed in 1983 with an initial extension of 124.5 hectares, over time more polygons were included until reaching the current extension of 237.3 hectares in 2005 (*Peralta & Prado, 2009*). However, the matrix surrounding the reserve (Ciudad Universitaria, Fig. 1, yellow line) has suffered fragmentation and habitat loss due to the development of university infrastructure over time. Despite the urbanization process, the reserve has managed to conserve the ecological heterogeneity and general conditions necessary to maintain a relatively similar number of bird species over time.

The proportion of species arriving and becoming extinct in the reserve was not influenced by time or by the species' residency status (Table 3). This implies that the proportion of birds arriving and becoming extinct to the reserve is similar for resident and migratory species (Fig. 3). This pattern makes sense since the effect of both ecological processes would cancel out, generating little variation in the number of species reported in the reserve over time. This pattern is consistent with the nestedness values reported in this study (Fig. 2, orange line). The arrival of resident and migratory birds to the reserve can be explained by the fact that in urban environments, birds will look for habitat patches that allow them to obtain resources and have optimal conditions (*Pacheco-Muñoz, Aguilar-Gómez & Schondube, 2022*). Because of this, urban green areas will concentrate a greater richness of bird species than completely urbanized areas. On the other hand, complex habitats are more vulnerable to disturbances and are prone to local extinctions (*Pimm, Jones & Diamond, 1988*). In the reserve, different disturbances could promote the loss of the heterogeneity of the microenvironments such as habitat loss and fragmentation, the introduction and establishment of exotic species that modify the structure and composition of the vegetation, and the presence of exotic predators such as dogs and cats.

We recorded the presence of 115 species to which we could not assign a residency category in the REPSA due to their low number of records. However, 68 of these species are considered residents in Mexico while 47 have some migratory movement within the country (*Berlanga et al., 2015*). These species contribute to the high turnover in community composition as they perform altitudinal or latitudinal migratory movements, but do not seem to use the reserve as a stopover or non-breeding area repeatedly or persistently over time. This pattern could reflect changes in the migration flyways that bird species follow to reach their non-breeding areas. Land-use change and infrastructure development promotes habitat loss and fragmentation along flyways and could limit the quantity and quality of stopover or non-breeding areas (*Kirby et al., 2008*). Additionally, climate change could affect the timing of migratory movements, as well as flight duration

due to changes in wind current speeds (*La Sorte et al., 2014*; *Barton & Sandercock, 2018*; *Howard et al., 2018*). The loss of suitable habitat could extend the duration of migratory journeys and increase the number of stopover areas that birds would need to complete their annual journey (*Howard et al., 2018*). The intensity of these adverse effects is associated with the distance each species migrates and the size of their non-breeding areas (*Barton & Sandercock, 2018*; *Patchett, Finch & Cresswell, 2018*). The high number of migratory species passing through the REPSA without their presence being reiterative over time could be a reflection of habitat degradation at a regional scale that forces species to modify their flyways in search of suitable areas to resupply and continue their migratory journey. However, species' migratory dynamics in their stopover and non-breeding areas in Mexico is understudied. This highlights the need to investigate these phenomena in order to contribute to the development of conservation strategies for North American birds.

Based on their persistence over time, we categorized 58 bird species as core, 33 as intermediate, and 167 as transient (Fig. 4, Table 2). The richness of core species will be related to local-scale environmental factors such as the resources and conditions of the studied area, while the richness of transient species will be influenced by factors at larger scales such as regional spatial heterogeneity or the potential number of species that may migrate through the studied area (*Magurran, 2007*; *Coyle, Hurlbert & White, 2013*). This implies that the persistence of core species could be an indicator that the conditions and resources that enable their persistence have been successfully conserved from 1973 to the present. Being immersed within Mexico City and being part of the central campus of the UNAM, the REPSA is subject to several disturbances such as habitat loss and fragmentation due to the development of university infrastructure (buildings and parking lots), inadequate management of solid and construction waste, the establishment of exotic plant species (*e.g.*, *Leonotis nepetifolia*) and the presence of exotic predators such as dogs and cats. However, we have implemented constant and permanent conservation actions to mitigate these pressures. For example, since 2015 we have carried out *L. nepetifolia* removal days to prevent its spread, and since 2011 we have maintained a constant monitoring program to prevent the establishment of feral dog and cat populations within the REPSA. In addition, we seek to conserve the patches of native habitat ("pedregales remanentes") that do not belong to the reserve but are essential for the functioning and connectivity of the habitat by promoting awareness activities for the university community. The REPSA also collaborates with Mexico City government institutions to improve the management of the system of natural protected areas and areas of environmental value and thus favor connectivity and conditions for urban wildlife in Mexico City.

On the other hand, the richness of transient species with intermediate persistence (Fig. 4, blue boxes) could represent a reservoir of potential colonizing species that could facilitate the maintenance of biodiversity and ecological functions in the future in the event that local environmental changes affect the persistence of core species (*Magurran & Henderson, 2003*). It is essential to maintain constant and permanent biological

monitoring that allows us to evaluate whether the rate of species turnover changes and if the persistence of the core species is maintained over time.

## CONCLUSIONS

In summary, our research provides the most complete historical checklist of bird species reported in the REPSA by covering a temporal period of more than 50 years and integrating data generated during this period from three sources: scientific literature, citizen science, and systematic biological monitoring. Turnover increased over time to values of 0.50 and showed an asymptotic trend over the 20-year census interval. Nestedness showed similar values over time. Additionally, we determine that the birds that arrive and become extinct in the reserve have similar proportions; this cancels out its effect and promotes maintaining similar species richness values over time. Finally, based on their persistence we were able to identify core, intermediate and transient species. Our results highlight the importance of the REPSA as a fundamental area for the avifauna of Mexico City and for a great variety of migratory bird species. This research is the baseline to continue and design new conservation strategies for the birds of REPSA and the environment on which they depend for example assessing population trends of the core species or determine their habitat use within the reserve.

## ACKNOWLEDGEMENTS

We thank the Secretaria Ejecutiva de la Reserva Ecológica del Pedregal de San Ángel for the facilities to carry out this research and the people and members of the Colaboradorus program who accompanied us to carry out the field observations. We also thank Stuart Pimm, and Gareth Russell and two anonymous reviewers for their suggestions that greatly improve our manuscript. Finally, we also thank Lynna M. Kiere for feedback on English language editing and manuscript proofreading.

### Funding
The authors received no funding for this work.

### Competing Interests
The authors declare that they have no competing interests.

### Author Contributions
- Ubaldo Márquez-Luna conceived and designed the experiments, performed the experiments, analyzed the data, prepared figures and/or tables, authored or reviewed drafts of the article, and approved the final draft.
- Pablo Arenas conceived and designed the experiments, performed the experiments, prepared figures and/or tables, authored or reviewed drafts of the article, and approved the final draft.

- M. Isabel Herrera-Juárez conceived and designed the experiments, performed the experiments, prepared figures and/or tables, authored or reviewed drafts of the article, and approved the final draft.
- Hilda Marcela Pérez-Escobedo conceived and designed the experiments, performed the experiments, authored or reviewed drafts of the article, and approved the final draft.
- Gustavo Hernández-Orta conceived and designed the experiments, performed the experiments, authored or reviewed drafts of the article, and approved the final draft.
- Guillermo Gil-Alarcón conceived and designed the experiments, authored or reviewed drafts of the article, and approved the final draft.

### Data Availability

The raw data is available in the Supplemental Files. The raw data shows abundance (number of individuals), persistence of each bird species recorded in the reserve and the composition species in each year analyzed.

Raw data are available at iNaturalist (https://mexico.inaturalist.org/observations?order=asc&place_id=112247&subview=map&taxon_id=3) and eBird (https://ebird.org/hotspots?env.minX=-99.3267509999999&env.minY=19.028231&env.maxX=-98.8529829999999&env.maxY=19.53966).

### Supplemental Information

Supplemental information for this article can be found online at http://dx.doi.org/10.7717/peerj.17888#supplemental-information.

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
