# Peer review of "Historic checklist, core species and temporal composition turnover of birds in an urban protected natural area in central México over 50 years"

_PeerJ, doi:10.7717/peerj.17888_

## Round 0.1 · original submission · Major Revisions

I disagree with reviewer 2, who recommends rejecting this. I think long-term studies of turnover are few, and what they tell us is important. That said, all three reviewers raise a long list of points that need attention. Please address all these issues in the revision that you return to us! The study by Russell et al is particularly relevant.

·

Basic reporting

Two methodological comments and a language one.

1) The authors test for a relationship between Jaccard dissimilarity (a mixture of turnover and nestedness) and census interval using a linear regression. But myself and others showed in 1995 (Russell et al. 1995, "Centuries of Turnover…") that the expected relationship between turnover and census interval is non-linear and asymptotic (and the same will be true for any dissimilarity metric in which turnover plays a large role). It is not clear that the data in this case are clearly non-linear, but it should be entertained and tested (it is basically the null hypothesis, given how turnover works), and if it is not non-linear as expected, that is worth discussion. (That paper also provides one reason why turnover might continue to rise, which is likely relevant to the current manuscript as well.)

2) In a related note, in that paper we included all pairwise comparisons between communities, rather than comparing each previous census only with the most recent. This provides more data points with which to examine the relationship between census interval and dissimilarity. I don't see why the authors could not do that here.

3) This is mainly a language note, but a phrase like 'the dissimilarity decreased significantly with time' is asking to be misinterpreted. It's not technically wrong in context, because by 'time' the authors are referring to dates prior to the most recent, compared to the most recent, but in most studies like this 'time' refers to the *interval* between censuses. Overall it is less ambiguous to just refer to census interval, in which case dissimilarity *increases* with it.

Experimental design

The paper is within scope. It doesn't fill an obvious knowledge gap, but historical datasets on community change are sufficiently rare that each one provides a valuable opportunity to replicate early studies. Other than the points above — which stem from a failure to show how this study does in many ways replicate an earlier one — the overall design and methods seem ok. (I put those methodological comments in the previous section because they refer to relevant previous work, but obviously they could be here as well.)

Validity of the findings

I can't say *for sure* how the validity might be affected by the methodical suggestions made earlier, but I am confident the main conclusion — that larger long-term dissimilarity is driven by ongoing changes the the environment — will not change.

Additional comments

Overall this paper uses a carefully curated historical dataset to make an interesting contribution to the literature on how and why community dissimilarly increases with time (interval).

·

Basic reporting

English used throughout manuscript is clear, but some minor inconsistencies were found. English language editing is suggested.

Experimental design

No comment

Validity of the findings

The data analysis is not statistically robust.

Additional comments

The focus of the manuscript is particularly relevant, mainly because the research involves the temporal turnover of avian species and a historic checklist of birds in a protected natural area immersed in a densely urbanized matrix. I consider that the information shown by the authors is important for the conservation of urban biodiversity. However, beyond this, I consider that the manuscript presents several issues that need to be considered by the authors.On the one hand, the study has a local focus (at least that is how the authors present it), so I suggest that the authors consider a local journal appropriate to their manuscript.

On the other hand, other more topics should also be addressed. Specifically, most of the introduction focuses on describing the Pedregal de San Ángel ecological reserve. I'm not saying it's bad, but the main issues of the study such as core species and composition turnover should be included in the introductory section. For example, lines 70-76 briefly describe the core species, but this topic should include theoretical information such as bibliographic literature, importance, and justification about the core species. I also did not find theoretical information regarding the species turnover. Are there studies on the turnover of bird species in urban ecological reserves? Beyond knowing whether migratory species continue to arrive at REPSA each year (as line 68 shows), are there important conservation implications of including temporal species turnover? The authors should be clearer about it.

- Include in the title the name of the country where the study was carried out.
- L80. The study has already been carried out, so the correct expression is "we evaluated"
- L131-132. An analysis of sampling effort should be performed. I think that if each site was visited twice per year, the number of species and their individuals are not representative values or each site has a low sampling effort, particularly if abundance was used to categorize the species, as well as for the linear regression using species abundance and persistence.
- L146-147. In Mexico, the fall-winter period for migratory birds is generally from September to February, although some migratory bird species can also be observed in March.
- L148. According to line 146, September is a month for winter migratory birds. Summer migratory birds are generally in Mexican territory from May to August.
- L164. Did the authors take into account the duplication of information? That is, the results of the theses or dissertations could have been published as scientific articles or book chapters. This could generate data duplication in the literature used. How did the authors avoid this?
- L229. I cannot catch how and why the authors infer a linear relationship among abundance and persistence, as well as among Jaccard's dissimilarity and time. The issue that concerns me is the type of data that was used in these analyses. That is, in the case of abundance, the number of individuals was used, which are absolute values; for persistence, the percentage of listings in which each species reported was used. So this analysis was performed from data with different nature, something I consider wrong. The authors also used linear regressions to test whether dissimilarity, turnover, and nestedness changed over time. Same issue, what type of data were used for this? Linear regression analysis has several assumptions such as linearity, normality and homoscedasticity. I suggest the authors review these assumptions before performing linear regressions. I think Figure 2 shows by itself how turnover and nestedness change over time. Also, you can include a significance analysis to validate the existence of significant differences in turnover and nestedness amog years.
- L259. Taking into account a similar comment aforementioned, I suggest the authors review specialized literature that allows them to complement their classification of bird species according to their migratory status. Most bird guides, particularly Howell and Webb (1995), include this information type.
- L265. Review genus of the House Finch.
- L357-358. The residence category was established based on the presence of each bird species during the bird surveys (lines 143-144). But the bird residence category from the number of records of the species is not mentioned. Along with this, lines 357-358 shows that 117 bird species do not have a residence category due to their low number of records. Does this “number of records” translate into frequencies, abundance or just the presence of a species? I suggest the authors complement the residency status of the birds based on specialized literature, as mentioned in line 359 where 64 species are migratory based on Berlanga et al. (2015). including this will allow for a clearer and more robust discussion of the results, particularly the temporal species turnover.
- L362-376. I suggest deleting this information. Instead, the authors should include information that discusses the implications of migratory birds on species turnover, using on the previous commentary.

Specific comments about the table 2:
- Table 2. Some inconsistencies were found in the nomenclature and systematic order of the bird species list. Review again and carefully the taxonomic proposal of Chesser et al. (2023).
- "Anseriformes" and "Anatidae" should be below the line, so this order and this family should be replaced by a title that includes orders, families and species.
- Mareca americana is the recent scientific name.
- Replace the genus with Sphyrapicus
- Why is Myiopsitta monachus classified as "A", but Agapornis personatus and A. roseicollis as "E" ? If all three bird species have been recorded in the study area, then they would have to be considered as exotic species, i.e. those are not native to a country or region, in this case Mexico. So I consider that the classification should be "E" for all three species. That is, species outside their geographical range and that are illegally traded, and whose presence in the reserve is likely due to accidental or deliberate releases. Also, Agapornis roseicollis belongs to the family Psittaculidae, which is located after the genus Amazona (Chesser et al., 2023). Include Agapornis personatus after A. roseicollis.

Reviewer 3 ·

Basic reporting

This study documents information on birds recorded over the last 50 years in the Pedregal de San Ángel Ecological Reserve. The authors performed an analysis of the species composition changes of the entire community and particularly with migratory birds. In addition, they identified the core species of the Reserve based on their persistence and related this variable to abundance (represented by the number of records obtained from the different information sources). The study is interesting, it seems to me that in general it is well written, although in my opinion the information contained has a more regional interest and lacks elements that could be of interest to readers from other parts of the world. Without detracting from the compilation, which seems to me to have represented hard work, it lacks analysis and testing of hypotheses that could make the manuscript more interesting.

Analyzing some variables related to urbanization to build ecological hypotheses based on species turnover data would have made a more interesting manuscript. I consider that an effort was made in the construction of the hypothesis of lines 84-87, by indicating that a positive relationship between persistence and abundance was expected due to the conditions and resources available in the Reserve. The point is that there is only a narrative without any evidence, since there is no data to confirm the “conditions and resources available” in the Reserve. Another weak point of the manuscript is the discussion section, which seems to me to be often repetitive of the results and does not clearly argue or support several of the findings obtained.

Experimental design

no comment

Validity of the findings

no comment

Additional comments

I include some minor comments that authors should be addressed
L79-80. I believe the hypothesis would have to be a little more specific. For example, what kind of changes are expected? in residents, in migratory, in certain groups of birds?
L104. Change “altitude” to “elevation”
L186. The fact that the same type of vegetation occurs in the reserve would not necessarily indicate the presence of the species.
L244-245. It is methodology and not results
L244-256. I consider that these lines should be better summarized and merged into a single paragraph.
L260.Eliminate an “e” in the word “escapees”
L316-317. This phrase should be merged with that in line 311 and 312
L320-325. This is not a discussion, again results are described. Authors should be more specific in constructing their ideas by arguing about the results, but not describing them again in this section.
L326-331. All this sentence is also results and not discussion.
L334-337. Again, this seems to be more speculation. I would suggest that they provide some evidence on possible effects of disturbance of these species in other geographical contexts and that their vulnerability to anthropogenic changes be demonstrated.
L360-362. I consider that it would have been useful to do the turnover analysis separately for the entire REPSA migratory bird community, in order to assess the contribution in terms of this turnover throughout the community.
L385-387. Again, this is result, not discussion.
L387-388. This phrase should be eliminated, it contributes nothing!
L410-415. It would be appropriate to mention some species and which ones, according to their requirements, phenotypic and adaptive plasticity, fulfill or could fulfill this function. Using the recorded species, a little more can help to exemplify and clarify the ideas even more.
L426-428. I suggest indicating some brief examples of what new conservation strategies could emerge from the results presented in this study.

---

## Round 0.2 · accepted · Accept

I have not heard from my previous reviewers about your revisions, so I have made this decision on my own after a careful read of your changes and responses to them. As I wrote before, I thought Reviewer 2 was wrong to recommend rejection — they had a long list of concerns, certainly, but none were fatal. The other reviewer made minor suggestions.